# Regulation of Microclimatic Conditions inside Native Beehives and Its Relationship with Climate in Southern Spain

**Sergio Gil-Lebrero [1],\*** 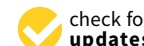**, Francisco Javier Navas González [2]**, **Victoria Gámiz López [1]**, **Francisco Javier Quiles Latorre [3]** **and José Manuel Flores Serrano [1],\***

1 Department of Zoology, University of Córdoba, 14071 Córdoba, Spain; victoriagamizlopez@gmail.com
2 Department of Genetics, University of Córdoba, 14071 Córdoba, Spain; v52nagof@uco.es
3 Department of Computer Architecture, Electrics & Electronics Technology, University of Córdoba, 14071 Córdoba, Spain; el1qulaf@uco.es
\* Correspondence: v72giles@uco.es (S.G.-L.); ba1flsej@uco.es (J.M.F.S.); Tel.: +34-957-21-8698 (J.M.F.S.)

**Abstract:** In this study, the Wbee Sensor System was used to record data from 10 Iberian beehives for two years in southern Spain. These data were used to identify potential conditioning climatic factors of the internal regulatory behavior of the hive and its weight. Categorical principal components analysis (CATPCA) was used to determine the minimum number of those factors able to capture the maximum percentage of variability in the data recorded. Then, categorical regression (CATREG) was used to select the factors that were linearly related to hive internal humidity, temperature and weight to issue predictive regression equations in Iberian bees. Average relative humidity values of 51.7% ± 10.4 and 54.2% ± 11.7 were reported for humidity in the brood nest and in the food area, while average temperatures were 34.3 °C ± 1.5 in the brood nest and 29.9 °C ± 5.8 in the food area. Average beehive weight was 38.2 kg ± 13.6. Some of our data, especially those related to humidity, contrast with previously published results for other studies about bees from Central and northern Europe. Conclusively, certain combinations of climatic factors may condition within hive humidity, temperature and hive weight. Southern Iberian honeybees' brood nest humidity regulatory capacity could be lower than brood nest thermoregulatory capacity, maintaining values close to 34 °C, even in dry conditions.

**Keywords:** *Apis mellifera iberiensis*; functional characterization; native subspecies conservation; climate change; sensors; temperature; humidity; weight; regulation

## 1. Introduction

The oldest references to the ability to regulate the temperature and relative humidity of bee colonies date back to the beginning of the 19th century. In 1806, Huber [1] reported that the range of temperature required for the development of the honeybee larvae would be around 35–36 degrees Celsius during summer in Switzerland. More than a century later, Oertel [2] reported unpublished work by E.F. Philips and Geo Demuth, who collected data of relative humidity within the winter cluster of bee colonies during the winter of 1923. Shortly afterwards, in the summer of 1926, Dunham measured beehive temperature hourly for a whole day [3], reporting an average temperature in the center of the brood nest of 34.1 °C. Although Oertel would be one of the first authors to measure humidity and temperature at different locations within the hives all those years later [2]. It would not be until the 90s, when the scientific attempts to the conjoined assessment of humidity and temperature would more frequently appear in the scene. In this context, the study by Atmowidjojo, et al. [4] established the tolerance of domestic and wild bees to heat and desiccation.

One decade after, the development of different real-time monitoring systems in beehives [5–8], has enabled improvement of knowledge regarding the idiosyncrasies of distinct bee colonies. For instance, the behavior of bees and their relationship with environmental conditions, their ability to adapt to environmental conditions depending on the type of hive in which the colony is housed, or the use of new systems to achieve greater profitability for beekeeping holdings (precision beekeeping) [9–16].

Precision beekeeping and beehive monitoring can report useful information for people associated with beekeeping to help manage their honeybee colonies. Benefits from it may occur first at the level of research, as large amounts of accurate and relevant information from the field can be collected. Second, at the level of average beekeepers, apiary handling may be optimized and visits to apiary and costs involved in travelling may be reduced which may improve the quality of production or prevent hive thefts. As drawbacks, the necessary equipment and its maintenance is expensive, even if costs are decreasing. Additionally, equipment installation may be problematic in remote areas, due to the need for electricity supply and internet access through the mobile networks [12].

Despite research on temperature and humidity regulation in bee subspecies and populations from Central Europe that can frequently be found, there is a patent lack of information in regard to bees in southern Europe. This situation contrasts the fact that not only more than 56% of the bee colonies registered in the European Union gather in this area, but also the same area accounts for the greatest professionalization of the sector and the highest production of honey [17].

Even if important intrabreed or interbreed genetic differences may occur, this unbalanced situation promotes the potentially incorrect application of benchmark values of northern populations to other populations that are distributed more meridionally [18,19]. Previous studies have shown how honeybee subspecies from the Iberian Peninsula (*Apis mellifera iberiensis*) may better adapt to the warm and dry climate of the area [8,20]. Therefore, it is essential to study the behavior of local honeybees from temperate zones in Europe, such as *A. m. iberiensis*. Increasing knowledge from these populations may imply great impacts from an ecological point of view, as it may promote the preservation of these subspecies and its genetic pool, but also from an economic and social perspective.

Iberian honeybee subspecies (*Apis mellifera iberiensis*) live in particularly sensitive bioclimatic areas. In this context, according to most climate models, global warming may have a greater impact on the climate of the Mediterranean area as longer, hotter and drier summers may progressively occur in future generations [21,22]. Even if public opinion has deemed this topic controversial, the consequences of global warming effects may already be noticeable. Rather than in isolation, with years falling outside the norm, an upward trend may be evidenced given that the current average annual temperature in Spain is 2 °C higher than it was fifty years ago [23]. For instance, 2017 was reported to be the warmest year recorded up until that point and the second driest in Spain in the 21st Century. During very hot and dry years, the amount of honey accumulated by bee colonies in spring considerably reduces. This not only affects the profitability of beekeeping farms, but also threatens the survival of the bees during summer, resulting in a period of food scarcity in southern areas [24,25]. Contextually, this study was developed in Córdoba (W77M+V5 Córdoba, Spain). According to the Spanish State Bureau of Meteorology, Córdoba is the city where the highest maximum temperatures recorded (46.6 °C) are reached and in which a greater number of days with average temperatures above 30 °C (119.7 days on average) occurred during the 1981–2010 climate period in Spain [26]. In addition, at a province level within Andalusia, Córdoba is the province where a greater temperature increase is expected in the coming decades in the Spanish territory [21]. Therefore, this location is especially sensitive to climate change and may be ideal for modelling the conditions which bees from other areas of the Mediterranean will confront in the future. The scarcity of scientific knowledge of the biology of local honeybee subspecies may only worsen the situation in southern Europe for the next few years. Concretely, this lack of knowledge may hinder the predictions for the potential evolution to expect from southern populations. Such an uncertainty makes it essential to reveal the adaptive mechanisms that Iberian bee colonies may implement. This is particularly important when the changes in the climatic conditions (certain combinations of climatic factors) in their area start to compromise the

normal development and health of honeybee colonies. Otherwise, in the likely event of an increase in the risks to which these populations are exposed, appropriate counteracting measures may not possibly be implemented.

If these events were to happen, the survival of local honeybee subspecies may imply the loss of ecological and genetic values that may be very useful in the future, provided the participation of domestic bees and other wild pollinators in pollination processes. A failure in the pollination process may put agricultural production and the entire ecosystem at risk. Collaterally, important economic and social consequences derived from the effects on honey, pollen and wax productions could arise.

Therefore, the first objective of the present study aims at providing benchmarks for internal temperature and humidity conditions of the local Iberian honeybee colonies using real-time monitoring; second, to determine how environmental conditions may affect the ability to regulate the temperature and humidity within the brood nest, food area and hive weight; third, to design regression models that may allow predicting the effects of different combinations of climatic factors on bee colonies' internal humidity and temperature and weight. The general aim of this study is to increase the knowledge and improve the understanding of the behavior of a highly productive local honeybee which is very well adapted to the Mediterranean climate. This may facilitate the implementation of conservation measures for the subspecies, protecting their role in Mediterranean ecosystems and the source of income of many families who depend on beekeeping.

## 2. Materials and Methods

### 2.1. Study Type and Study Units

The present observational cohort study considered individual bee colonies as study units [26,27]. These bee colonies belonged to the local populations of Iberian subspecies (*Apis mellifera iberiensis*) and were monitored for two complete beekeeping seasons, from 31 March 2016 to 26 February 2018 (from early spring to late winter).

### 2.2. Hive Type, Location and Spatial Configuration

Bee colonies were housed in Langstroth® hives that were placed on 50 cm high metal supports. Hives were spatially configured forming a row which was oriented to the south. The whole row consisted of 20 beehives from which only 10 were monitored for internal humidity, temperature and weight. The hives selected for monitoring were located in the western end of the row. When hives are distributed forming rows, foragers tend to concentrate at the ends when they return to their hive as a result of drift or confusion [28]. Spatial configuration of the hives is depicted in Figure 1.

Beehives were located in the experimental apiary of the University of Córdoba (37°55′34.37″ N, 4°43′25.60″ W). Córdoba's climate is defined as a sub-continental Mediterranean climate [21], characterized by moderately cold winters, and very hot and arid summers. The apiary is located in a transition zone between the countryside and the mountain, and therefore is surrounded by large dryland agricultural areas (where cereal, sunflower, and olive crops predominate, with pastures destined to livestock feeding) which become progressively denser Mediterranean-type forests (with *Querqus* spp., *Cystus* spp. and small forests of *Eucalyptus* spp. among others).

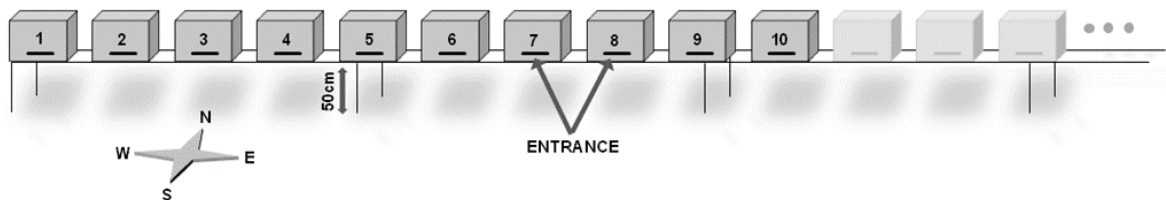

**Figure 1.** Hives spatial configuration. The number of each hive corresponds to its position in the row.

### 2.3. Colony Creation, Maintenance and Monitoring

As a common feature in cohort studies [29], a non-interventionist approach was followed and experimental units (i.e., colonies) were not manipulated apart from regular handling. Regular handling consisted of the tasks needed for the harvest of honey, the supply of maintenance feed in times of lack of food (in our case during August and between January and February) offering *ad libitum* feed in tablets of water with sucrose, and mandatory treatments in autumn for the control of the *Varroa destructor* mite according to the Spanish regulation. Amitraz acaricide (Amicel Varroa®) was applied following the directions for use of the product. Treatments were administered from 19 September 2016 to 24 October 2016 and from 3 October to 7 November 2017.

At the end of spring, supers were added to adapt the size of the beehive to that of the colonies that needed it. A honey super is a part of a commercial or other managed beehive that is used to collect honey. Supers are extensions of the hive (box), which are placed on top of the main body, allowing the colony of bees to have more space to develop at the times of the year when they need it. Other common practices in beekeeping, such as the reinforcement of weak colonies with breeding and adult population of the strongest colonies, were not performed so as to fulfill the non-interventionist policy of the study.

All colonies were monitored in their natural setting and underwent the same experimental protocol. Bee colonies were obtained from nucleus ("nucs") colonies of 5 frames full of local populations of honeybees, in good sanitary conditions and with sister queens artificially reared before the study, free mated and with the presence of eggs from the beginning.

Only two colonies remained active during the entire period of the study. The rest of colonies needed to be replaced at some point due to different reasons, such as colony mortality, weakening, the presence of important pathologies such as *Ascophaera apis* or *Varroa destructor* or the infestation by *Galleria mellonella* larvae. Thirteen colony replacements were performed during the two years of study. Percentage of colony replacement was 52% (13 out of each 25 colonies), due to mortality or weakness of colonies. Such replacement and the introduction of new colonies were necessary to ensure a sufficient number of workers were present in each beehive under study, hence the ability of the colony to maintain normal homeostasis within the hives was not affected. All colonies that needed to be replaced were substituted by new nucleus colonies of 5 frames with normally functioning queens. Considering colony replacements, a total of 25 colonies, randomly named from "A" to "Y", were used for the whole study. Twelve colonies were still alive at the end of the study.

### 2.4. Data Collection Computerized System

Wbee Sensor System was used to record internal humidity, temperature and weight data from beehives. Wbee consists of a wireless electronic device specifically developed for monitoring honeybee colonies [8]. This hierarchical system works at three levels, which guarantees data security and allows real-time remote access to the data from anywhere. A set of sensors controlled by a Waspmote®-based processor is placed within each beehive, and periodic reviews were performed to ensure that the sensor was within the proper area within the hive, repositioning sensors in case it was necessary. These processors follow the orders of a central computer located in the apiary that communicates with them via wireless technology. This central computer is responsible for the coordination and integration of the information collected from the beehives. Once the data have been collected, the central computer forwards the data to a server which records it, stores it and grants later access to all data. This system monitors the temperature (°C) and relative humidity (%) in different areas within the beehive, as well as its total weight (kg) in real-time. For this study, humidity and temperature sensors were located in the center of the brood nest (humidity 1 and temperature 1) and in the food zone (humidity 2 and temperature 2).

Environmental climatic data were obtained from the weather station of the university and placed one kilometer away from the apiary. Hourly climatic records for average temperature (°C) (Mean Ext Temp), minimum temperature (°C) (Min Ext Temp), maximum temperature (°C) (Max Ext Temp), relative humidity (%) (Ext Humidity), solar radiation (W/m$^2$) (Radiation), atmospheric pressure (hPa)

(Pressure), precipitation (mm) (Rainfall) and average wind speed (m/s) (Wind), were considered in this study.

Collaterally, potential conditioning factors that could affect internal humidity and temperature and weight were considered as well. These factor were as follows; the year and months in which the study took place, the hour at which each observation was made, position (setting the physical position of the hive in the hive row, with 1 corresponding to the end of the row and 10 to the hive which locates closer to the center in the row) (see Figure 1), colony (understood as the superorganism or the set of worker and queen bees capable of maintaining the homeostasis of the colony and its population over time, within normal seasonal fluctuations) and supers (between 0 and 2 supers of Langstroth model, depending on the time of the year).

*2.5. Statistical Analysis*

2.5.1. Within Hive Temperature, Humidity and Weight Benchmark Values

All the data obtained from the Wbee Sensor System (n = 545,122 total observations) were tested for common parametric assumptions to determine the most appropriate statistical methods to use in the contexts of the properties of our data. Following the guidelines presented in Laerd Statistics [30], the Friedman test was used, as related samples (one sample of participants (hives) with repeated measures per hive) were considered in this study [31], to test for differences in the median of the continuous dependent variables of humidity 1 (brood nest) and 2 (food area), temperature 1 (brood nest) and 2 (food area) and weight between factor categories as data violated parametric assumptions (non-normality or heteroscedasticity). As suggested by the same guidelines, median values for each of the related groups should also be reported. However, at this stage, we only know that there are differences somewhere between the related groups, but we do not know exactly where those differences lie.

To examine where the differences actually occur, we need to run separate Wilcoxon signed-rank tests on the different combinations of related groups. Additionally, we need to use a Bonferroni adjustment on the results you get from the Wilcoxon tests because we are making multiple comparisons, which makes it more likely that we will declare a result significant when you should not (a Type I error, that is the rejection of a true null hypothesis, also known as a "false positive" finding or conclusion). The Bonferroni correction is calculated by taking the significance level (*p*-value) that was initially considered and dividing it by the number of tests performed. Related-samples Friedman's two-way analysis of variance by ranks and Wilcoxon signed-rank tests were performed using the nonparametric package of the related samples procedure of SPSS Statistics for Windows, version 25.0, IBM Corp. (2017) [32].

2.5.2. Effects of Conditioning Factors on Hive Temperature, Humidity and Weight Evolution

After median benchmark values for internal humidity, temperature and weight had been determined, an analysis of the effect size or the power of potential conditioning factors was performed prior to deciding which set of risk factors should be considered to build evaluation models. The evaluation of effect size or power has been suggested as the preferable approach to perform prior to linear regression [33].

As the variables of humidity 1 and 2, temperature 1 and 2 and weight did not fit to a normal distribution ($P < 0.001$), nor were they homoscedastic, a Mann–Whitney U test was used to detect a significant effect for the year factor, considering only two possible years, 2016 and 2017. Simultaneously, a Kruskal–Wallis H test was performed to detect significant effects of factors for which more than two possible levels existed. We show a summary of the levels for all the factors included in the model in Table 1. Mann–Whitney U and Kruskal–Wallis H tests only detect significant effects of factors through the comparison of the distributions of a certain variable across the levels of independent factors, but do not tell you between which levels such exact difference occurs. Hence, Dunn's test was

used to identify the particular pairs of levels of each factor between which distribution of humidity 1 and 2, temperature 1 and 2 and weight was significantly different. Bonferroni correction method has been reported to control the occurrence of family-wise errors (FWE), that is the chance of one or more false positives [30]. False positives are likely to reflect biases such as those derived from confounding factors [31]. Conclusively, Bonferroni correction was used as a part of the procedures aiming at minimizing or controlling confounding effects [33]. All nonparametric tests were carried out using the independent samples package from the non-parametrical task of SPSS Statistics for Windows, version 25.0, IBM Corp. (2017).

**Table 1.** Summary of the factors, levels for all the factors, the tests used to detect significant differences in the distribution of variables across factor levels and the tests to measure for the effect size of each factor on each variable.

| Factor | Level | Test | Effect Size |
|---|---|---|---|
| Year | 2016–2017 | Mann–Whitney U | Rank biserial correlation (*r*) |
| Month | January to December | Kruskal–Wallis H | Cohen's f |
| Hour | 24 h | Kruskal–Wallis H | Cohen's f |
| Position | 1 to 10 | Kruskal–Wallis H | Cohen's f |
| Colony | A–Y (25 colonies) | Kruskal–Wallis H | Cohen's f |
| Supers | 0, 1 and 2 supers | Kruskal–Wallis H | Cohen's f |
| Wind (m/s) | 0–12.4 | Pearson correlation analysis | Pearson's *r* |
| Min Ext Temp (°C) | −2.6–46.4 | Pearson correlation analysis | Pearson's *r* |
| Max Ext Temp (°C) | −0.8–47.3 | Pearson correlation analysis | Pearson's *r* |
| Mean Ext Temp (°C) | −1.8–46.8 | Pearson correlation analysis | Pearson's *r* |
| Ext Humidity (%) | 16–100 | Pearson correlation analysis | Pearson's *r* |
| Radiation (W/m$^2$) | 0–323 | Pearson correlation analysis | Pearson's *r* |
| Pressure (hPa) | 613–1033 | Pearson correlation analysis | Pearson's *r* |
| Rainfall (mm) | 0–11.5 | Pearson correlation analysis | Pearson's *r* |

After conducting the Mann–Whitney U test (for two levels) and Kruskal–Wallis H with three or more levels (k), the effect size or power of each factor on humidity and temperature 1 (brood nest) and humidity and temperature 2 (food area) and hive weight was computed [34]. Rank biserial correlation *r* [35] and Cohen's f were used as quantification measures. Contextually, Cohen's f allows analysis of the relationship between a continuous or ordinal variable and a categorical variable in the case when the latter has more than two possible levels (k values). SPSS cannot calculate Cohen's f directly, but it can be calculated through SPSS outputs for partial eta$^2$ ($\eta p^2$) [36]. Additionally, Pearson's correlation coefficient was used to measure for the relationship between pairs of continuous variables and covariates (Figure 2). Cohen's guidelines for *r* are that a small effect is 0.1, a medium effect is 0.3, and a large effect is 0.5 [37]. Guidelines to interpret Pearson's correlation coefficients can be found in Profillidis and Botzoris [38]. Effect size or power was calculated using the Crosstabs procedure from SPSS Statistics for Windows, version 25.0, IBM Corp. (2017) (Figure 2) [32]. The Pearson correlation analysis was performed using the Correlations procedure of SPSS Statistics for Windows, version 24.0, IBM Corp. (2016).

2.5.3. Model Complexity Dimensionality Reduction and Predictive Regression Models

Once the factors and covariates with significant effects had been identified, a categorical principal components analysis (CATPCA) was used to reduce the dimensionality of the data and prevent Type I errors (false positives) with the Optimal Scaling procedure from the Dimension reduction task from SPSS Statistics for Windows, version 24.0, IBM Corp. (2016). Dimensionality reduction was performed through the construction of principal components (PCs). PCs are new variables that are constructed as linear combinations or mixtures of the original factors. These combinations are performed in such a way that the PCs are uncorrelated and most of the information within the original factors is compressed into the first components that are constructed. Organizing information in PCs in this way allows

for reduction in the number of factors considered (dimensionality) without losing much information. Afterwards, the components with low information will be discarded and the remaining components will be considered as our new variables.

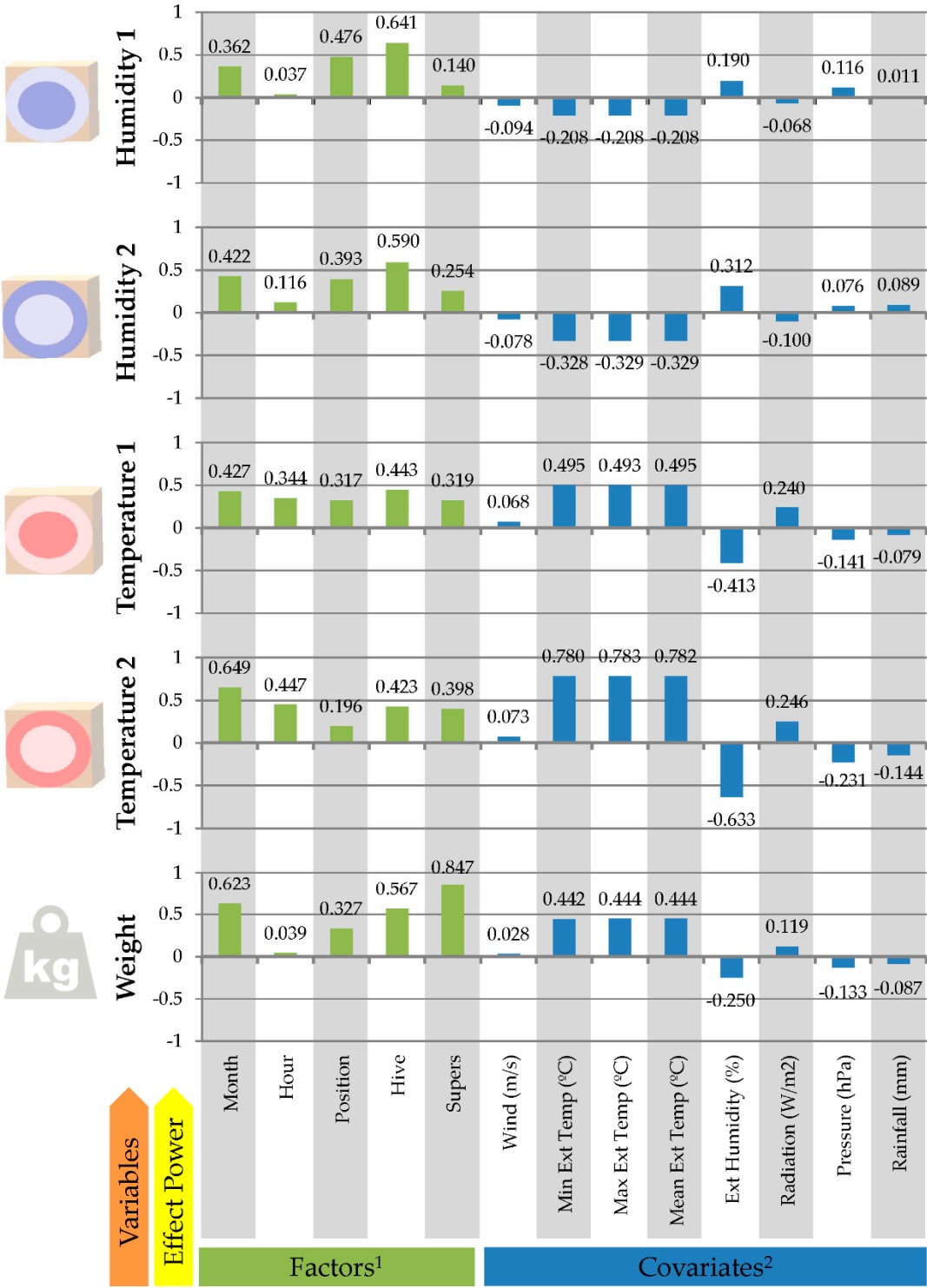

**Figure 2.** Summary of the effect size of conditioning factors on the variables of brood nest temperature and humidity (Temperature and Humidity 1), food area temperature and humidity (Temperature and Humidity 2) and hive weight using Cohen's f, and Pearson correlation coefficients between the same variables and environmental continuous covariates (2) in studied bee colonies (n = 545,122).

After PCs have been constructed, the next step is to identify which of the original factors on each PC are able to explain the highest possible percentage of variance in our dataset (Table 2). In the CATPCA, only those variables whose loading coefficients were greater than or equal to |0.5| were considered predictive variables in the categorical regression model [39]. The rest were considered significantly confounding factors and were discarded (Table 3). High component loadings may be suggestive of a common factor that significantly correlates with all items considered [40]. All those factors and variables that present values lower than |0.5| will be good candidates to discard from our model given their confounding nature.

To test whether dimensionality reduction was performed properly, CATPCA reliability must be tested. To test the reliability of CATPCA, we used the parameter known as Cronbach's alpha. When the original factors considered in CATPCA are indeed each of the levels of the same scale, Cronbach's alpha allows for estimation of the internal consistency, reliability and ability of that scale to measure for the same variable, construct or theoretical dimension [41]. However, when we aim to identify and quantify the effects of independent factors on a certain variable (not the levels in the same scale), Cronbach's alpha can be used to quantify unidimensionality of each PC, block or subset of factors. In these contexts, unidimensionality is the quality of such factors to represent the common latent variable, construct or theoretical dimension [42].

As a general criterion, George and Mallery [43] suggested the following recommendations for evaluating unidimensionality and/or scale reliability after Cronbach's alpha coefficients: >0.9 is excellent, >0.8 is good, >0.7 is acceptable, >0.6 is questionable, >0.5 is poor and <0.5 is unacceptable. These cut off values account for the ability of a certain PC, block or subset of factors (individual Cronbach's alpha for each of the PCs) or of PCs, blocks or subsets on the whole (total Cronbach's alpha) to measure for a single variable, construct or theoretical dimension. However, this rule of thumb may derive from a misinterpretation of Nunnally and Bernstein [44], who did not report any cut off values, but reported different contexts for Cronbach alpha acceptance leaving limit values very open.

However, Cronbach's alpha and percentage of explained variance must not be interpreted separately, but simultaneously considering the Kaiser criterion, which provides a measure of the balance between the percentage of explained variance and reliability [45]. Commonly, Kaiser's rule that is applied states that all PCs with eigenvalues lower than 1 should be discarded from the analysis. The rationale behind the Kaiser criterion is that eigenvalues less than 1 account for less of the total variance than do any one of the original variables. However, in some cases, it is desirable to select a minimum value lower than 1 to allow for sampling variation and a more liberal eigenvalue threshold [46]. Jollife [47] suggests lowering the threshold to retain any PC with an eigenvalue over 0.7.

Although principal components analysis (PCA) can be used as a good dimensionality reduction tool that permits us to identify the factors that contain the most information in the dataset [48], it should not be used as a variable selection tool. At this point, an important thing to realize is that, as opposed to original factors, PCs are less easily interpretable and do not have any practical meaning since they are constructed as the linear combinations of the original factors that each PC, subset or block comprises [49]. This drawback also applies to complete principal component regression (PCR) which consists of the use of PCs as the factors in regression analysis, after which variable selection regression procedures are applied to determine the components to retain. Additionally, the decision about how many principal components to keep in PCR analysis does not depend on the response variable [50]. Consequently, some of the factors kept might not be strong predictors of the response, while some of them that you drop might be excellent predictors. Consequently, when the primary goal is variable reduction, then variable selection procedures on original variables should be used rather on the principal components (PCs).

The use of PCs as inputs has been reported to improve multiple linear regression model predictions by reducing their complexity and eliminating data collinearity, that is if reduced subsets of original factors are used [51]. In this regard, categorical regression (CATREG) analysis identifies which of the factors in the previously reduced components have a significant relationship with the dependent

variables that are regressed. In this context, PCA has been reported to provide exact solutions for dependent variables if the linear regression approach is used in PCA follow-up analysis as a factor selection tool after dimensionality reduction practices have been implemented [52].

We used linear regression with a forward stepwise selection (FSWS) approach [53,54]. FSWS only requires p models to choose the best fitting model and can deal with tasks where $p > n$ (it simply adds a stopping rule when $p = n$), with p being the number of predictors/factors and n the number of observations. CATREG uses optimal scaling procedures to determine the linear relationship between a dependent variable and one or more independent factors which can be measured using different units. It yields an optimal linear regression equation describing how changes in factors affect units of standard deviation of the dependent variable, hence evaluates the relationship between independent factors and the dependent variables and measures for their explicative or predictive conjoined potential.

Following the common notation models, the regression model for each predicted variable was, $Y_m = \beta_n Z_n + \varepsilon$, where Yn is the $m^{th}$ dependent variable, βn is the regression coefficient for the $n^{th}$ factor significantly loading ($\geq |0.5|$) within the $n^{th}$ principal component, $Z_n$ is the score obtained in the field for the $n^{th}$ factor and $\varepsilon$ represents the estimation error. Likewise, to estimate the prediction error of the categorical regression model, we used the cross-validation resampling technique of "k" (k-fold cross-validation), since it allows us to evaluate models in which several predictors are used and is more adequate than other models of resampling such as bootstrap 0.632, given our sample size and a high number of predictors [55].

Then, to determine the capacity of the model designed to fit the data in our population, we evaluated R-squared. R-squared is also called the coefficient of determination, or the coefficient of multiple determination for multiple regression. For the same dataset, higher R-squared values represent smaller differences between the observed data and the fitted values. R-squared is the percentage of the dependent variable variation that a linear model explains. As the independent factors considered were categorical and the data were sorted into categories following different criteria, we used standardized coefficients to interpret and compare their effects on our dependent variables, avoiding the potential distortion derived from factors being measured in different units. Additionally, values of $R^2$ can be very important in the justification of the efficiency and accuracy of PCA if linear regression is used in the follow-up analysis. If $R^2_k$ (determination coefficient of reduced model) is close to the $R^2$ value of the model with all of the explanatory variables, then the present model is adequate; otherwise we need to increase k (number of variables) such that $R^2_k$ comes closer to complete the model's $R^2$ [52].

**Table 2.** Summary of the results of principal components reliability/unidimensionality analysis.

| Principal Components, Blocks or Subsets | Cronbach's Alpha | Eigenvalues |
|:---:|:---:|:---:|
| PC1 | 0.870 | 5.193 |
| PC2 | 0.634 | 2.434 |
| PC3 | 0.449 | 1.715 |
| PC4 | 0.263 | 1.323 |
| PC5 | −0.158 | 0.872 |
| Total | 0.984 | 11.537 |

**Table 3.** Loading for each component included in the model for each of the dimensions stipulated in the categorical principal components analysis (CATPCA).

| Dimension / Factor | Principal Components, Blocks or Subsets | | | | |
|:---:|:---:|:---:|:---:|:---:|:---:|
| | 1 | 2 | 3 | 4 | 5 |
| Year | −0.325 | −0.027 | **0.595** | −0.319 | −0.445 |
| Month | 0.489 | 0.120 | **−0.655** [1] | −0.025 | 0.133 |
| Hour | **0.620** [1] | −0.068 | **0.568** | −0.001 | 0.139 |
| Position | 0.128 | **0.988** [1] | −0.032 | 0.015 | 0.054 |
| Colony | 0.120 | **1.002** [1] | −0.011 | 0.005 | 0.036 |

**Table 3.** *Cont.*

| Factor / Dimension | Principal Components, Blocks or Subsets | | | | |
|---|---|---|---|---|---|
| | **1** | **2** | **3** | **4** | **5** |
| Supers | 0.148 | **−0.652** [1] | −0.331 | 0.068 | 0.146 |
| Wind (m/s) | 0.439 | −0.061 | **0.515** | 0.261 | 0.271 |
| Min Ext Temp (°C) | **0.992** [1] | −0.028 | −0.107 | −0.009 | −0.086 |
| Max Ext Temp (°C) | **0.996** [1] | −0.028 | −0.106 | −0.024 | −0.091 |
| Mean Ext Temp (°C) | **0.997** [1] | −0.029 | −0.106 | −0.016 | −0.092 |
| Ext Humidity (%) | **−0.901** [1] | 0.047 | −0.165 | 0.169 | 0.143 |
| Radiation (W/m$^2$) | **0.552** | −0.036 | 0.371 | −0.125 | 0.314 |
| Pressure (hPa) | −0.297 | −0.009 | 0.021 | **−0.721** [1] | **0.570** |
| Rainfall (mm) | −0.211 | 0.019 | 0.191 | **0.764** [1] | 0.263 |

In bold: significantly loaded components ≥|0.5|. [1] Predictive variables in the categorical regression (CATREG) model.

## 3. Results

Data were non-normally distributed and heteroscedastic (Shapiro–Francia and Levene's tests $p < 0.05$, respectively). As a result, and although no outlier was detected following the methods proposed by Hoaglin and Iglewicz [56], and no multicollinearity problem could be presumed (as "variance inflation factor" or VIF values were <10 [57,58]), a nonparametric approach was suggested.

### 3.1. Within Hive Temperature, Humidity and Weight Benchmark Values for Iberian Honeybees

A summary of the descriptive statistics for internal environmental hive and hive weight variables and external environmental covariates in Iberian bee colonies can be seen in Figure 3. Average relative humidity values of 51.7% ± 10.4 are reported for humidity in the brood nest (Humidity 1), while average humidity in the food area (periphery of the colony, Humidity 2) was 54.2% ± 11.7. Maximum values for humidity in the brood nest and food area were 91.3% and 92.6%, respectively, while minimum values in both cases were 15%. Average temperature in the brood nest was 34.3 °C ± 1.5 (Temperature 1) while average temperature in the food area was 29.9 °C ± 5.8 (Temperature 2), with minimum and maximum temperatures ranging from 29.5 to 45.9 °C in the first case, and from 4.4 to 49.8 °C in the second, respectively. Average beehive weight was 38.2 kg ± 13.6, with values that ranged from 16.1 kg to 75.0 kg. Friedman test reported significant differences ($P < 0.05$) in median values for temperature and humidity across sensor positions (brood nest and food area) and external data as shown in Table 4. Climatic data obtained during the study period reported average values of environmental relative humidity of 56.4%, with a maximum of 100.0% and a minimum of 16.0%. Contrastingly, for the same period, average environmental temperature was 25.9 °C, with values ranging between the maximum value of 47.3 °C and the minimum value of −2.6 °C.

**Table 4.** Results for the related-samples Friedman's two-way analysis of variance by ranks to detect differences across external, peripheral and center temperature and humidity.

| Pairwise Comparisons | Test Statistic | Degree of Freedom | Asymptotic *p*-Values (2-Sided Test) |
|---|---|---|---|
| Mean Ext Temp, Temperature 2 | 121,936.97 | 1 | 0.001 |
| Mean Ext Temp, Temperature 1 | 227,059.06 | 1 | 0.001 |
| Temperature 1, Temperature 2 | 351,630.06 | 1 | 0.001 |
| Ext Humidity, Humidity 2 | 68.43 | | 0.001 |
| Ext Humidity, Humidity 1 | 1399.96 | 1 | 0.001 |
| Humidity 1, Humidity 2 | 21,455.82 | 1 | 0.001 |

Supplementary Table S1 shows the different pairs of levels of each factor, between which a significant difference in distribution was found. Afterward, Supplementary Table S2, as suggested by the guidelines in Laerd Statistics [30], presents detailed information of the median benchmark values of Humidity 1 and Temperature 1 (brood nest), Humidity 2 and Temperature 2 (food area) and hive weight for each level of the aforementioned factors of Year, Month, Hour, Position, Colony and Supers. Dunn's test suggested that the median benchmark temperatures recorded in 2017 were significantly higher than values in 2016. The opposite circumstances were described for humidity, with 2016 reporting higher median benchmark values. There were significant differences across all months for almost all the variables measured except for Weight (between January–December, December–March, March–April, and February–October), and humidity in brood nest or Humidity 1 (between August–September, May–October, January–December, and December–February).

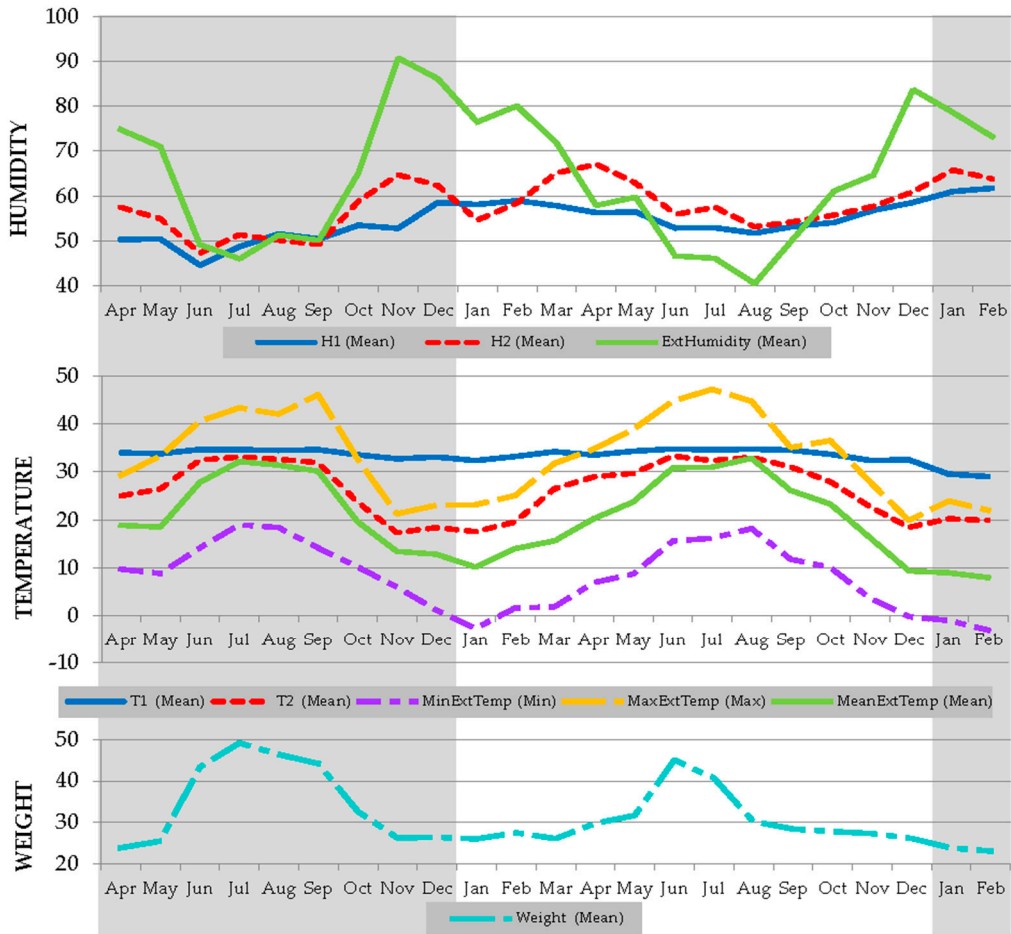

**Figure 3.** Seasonal variation in minimum, maximum, and mean external temperature and humidity, brood nest temperature (T1) and humidity (H1), food zone temperature (T2) and humidity (H2) and weight during the period of study.

All variables reported significantly different values at almost all hours except for two time slots which had an opposite progression along the day. Both slots comprised 4 to 5 h during which the variables were not significantly different, that is brood nest and food area humidity and temperature and hive weight constantly decreased or increased each hour at intervals of 0.1 to 0.3 units (°C for temperature, % for humidity, and kg for weight). Such progressive decreasing or increasing intervals occurred towards the ends of each slot (Supplementary Tables S1 and S2).

Significantly different benchmark values were reported depending on the position of the hive within the row and the number of supers used. The highest values for the median were reported

for positions from 3 to 5 and 4 to 6 when humidity 1 and humidity 2 were considered, respectively. In addition, 1 to 3 and 4 and 8 positions within the row for temperature 1 and temperature 2 were reported, respectively, while Weight reported the highest medians for 2 and 7 positions. All variables significantly differed depending on the supers considered (Supplementary Tables S1 and S2) with 0 reporting the highest medians (53.8 and 50.1 for Humidity 1 and Humidity 2, respectively) for humidity values and 2 for temperatures (35 and 32.8, for temperature 1 and 2, respectively) and weight (59.5).

### 3.2. Effects of Conditioning Factors on Hive Temperature, Humidity and Weight Evolution

Figure 2 reports the results of the effect size analysis for all the factors and covariates on all the variables tested. The lowest effect size, hence variable relationship, was reported for the Hour factor on Humidity 1 and Weight variables (Cohen's f: 0.037 and 0.039, respectively), while the highest value was reported for Supers factor on Weight (Cohen's f: 0.847). Month and Colony effect sizes in all the variables were moderately high when compared to the rest of the effect sizes, whose ranges were more widely variable. Temperatures 1 and 2 were moderately affected by the Hour, while Humidity 1 and 2 reported similar but slightly lower effect sizes for the factors of hive Position within the row and Supers. Colony, Month and Supers factors reported the highest effect sizes on Weight (Cohen's f: 0.567 and 0.847 for Colony and Supers, respectively).

### 3.3. Model Complexity Dimensionality Reduction

Once significantly non-confounding factors had been identified, dimensionality reduction was implemented using CATPCA to retain the minimum number of factors that were able to describe the highest percentage of variability in our dataset to reduce the complexity of future models. The five-dimensional model results are shown in Table 2. Only 11 of the elements studied contributed to the five-dimensional model in a meaningful way (factor loadings ≤ |0.5|, Table 3). The different components (PC1, PC2, PC3, PC4 and PC5) were best described by the factors highlighted in bold in Table 3 (factor loadings ≥ |0.5|).

The first component (PC1) comprised positive significant positive loadings for Hour, Min Ext Temp, Max Ext Temp, Mean Ext Temp and negative significant loading of Ext Humidity. Hence, PC1 could be called thermal sensation time variation and supports the results addressed by the Dunn Test Pairwise comparisons regarding temperature and humidity time slots previously described. The second component (PC2) comprises the factors of position, colony, and supers, which define this component as spatial location and configuration of the colony. Position and Colony loaded positively and highly while Supers presented negative, moderately high loading, which also supports the results reported by the Dunn test for the absence of supers when compared to the presence of one or two of them. The third principal component (PC3) comprised negative, moderately high loading of the month, which represents the annual cycle. A fourth principal component (PC4) was reported clustering Pressure and Rainfall, which highly negatively and positively loaded, respectively, with almost the same absolute value. This PC4 involves the reactions of the beehives to conditions related to weather depression moments. The fifth component (PC5) only comprised pressure as a significantly loaded factor.

When independent factors are considered, low PC values may suggest a lack of homogeneity of these factors [59], which may likely occur when the factors measured using different units are considered to determined their conjoined effect over certain variables, constructs or theoretical dimensions (temperature and humidity at different points in the hive and weight, in our case). In these situations, component loadings may turn negative, as indicative of an inverse effect of a factor on a certain variable, construct or theoretical dimension.

In these contexts, low-negative Cronbach's alpha for a certain PC, block or subset may be acceptable if the factors it comprises do not correlate with the factors included in the rest of the PCs, blocks or subsets [60]. Although PC5 did not reach generally acceptable levels for Kaiser criterion (>1 eigenvalues), its exclusion may not make any difference provided the variable of pressure

(only significantly loaded variable for PC5) also significantly loaded for PC4, hence, should be retained in the analysis anyway. Additionally, the very weak to lacking range of correlations found (−0.19 to 0.01) [37] between pressure or rainfall and the rest of the factors may statistically support the inclusion of PC4 and 5 (only significantly loading factors in PC4 and 5 for which Cronbach's alpha was below 0.7, respectively).

### 3.4. Predictive Regression Models

After CATPCA, all significantly non-confounding factors (factor loadings ≥ |0.5|, Table 3) were considered predictive factors in the categorical regression model [39]. Year, wind and radiation were detected as significantly confounding elements (factor loadings < |0.5|, Table 3), thus, were not considered for the CATREG analysis. Since the stepwise method was used, no multicollinearity problem could be assumed, as it had also been previously supported by the lower than 10 values of VIF. Supplementary Table S3 shows a summary of the results of the CATREG analysis and the significance level of each of the factors considered after dimensionality reduction.

**Table 5.** Model summary and standardized solutions for the regression equations for brood nest and food area humidity and temperature (humidity and temperature 1 and 2, respectively) and hive weight.

| General Model Regression Equation | Legend |
|---|---|
| $Z'y_{h1h2t1t2w} = \beta_{month} * Z_{month} + \beta_{hour} * Z_{hour} + \beta_{position} * Z_{position} + \beta_{colony} * Z_{colony} + \beta_{supers} * Z_{supers} + \beta_{minexttemp} * Z_{minexttemp} + \beta_{maxexttemp} * Z_{maxexttemp} + \beta_{meanexttemp} * Z_{meanexttemp} + \beta_{exthumidity} * Z_{exthumidity} + \beta_{pressure} * Z_{pressure} + \beta_{rainfall} * Z_{rainfall}$ | $Z'y_{h1h2t1t2w}$ = Z score for each variable (humidity 1, humidity 3, temperature 1, temperature 3, weight). |
| | $\beta$ = standardized coefficient for each of the factors appearing in the subindex. |
| | Z = Z scores for each of the factors appearing in the subindex. |

| Specific Regression Equations | | Legend |
|---|---|---|
| | | $Z'y_{h1}$ = Z score for Humidity 1 variable. |
| | | $\beta_{month}Z_{month} = 0.214 (Z_{month})$ |
| | | $\beta_{hour}Z_{hour} = -0.021 (Z_{hour})$ |
| **Humidity 1 (%). Brood nest** | $Z'y_{h1} = 0.214(Z_{month}) - 0.021(Z_{hour}) + 0.560 (Z_{position}) + 0.289(Z_{colony}) + 0.053(Z_{supers}) + 0.028(Z_{minexttemp}) + 0.110(Z_{maxexttemp}) + 0.039(Z_{meanexttemp}) + 0.192(Z_{exthumidity}) - 22120.007(Z_{pressure}) - 0.007(Z_{rainfall})$ | $\beta_{position}Z_{position} = 0.560 (Z_{position})$ |
| | | $\beta_{colony}Z_{colony} = 0.289 (Z_{colony})$ |
| | | $\beta_{supers}Z_{supers} = 0.053 (Z_{supers})$ |
| | | $\beta_{minexttemp}Z_{minexttemp} = 0.028 (Z_{minexttemp})$ |
| | | $\beta_{maxexttemp}Z_{maxexttemp} = 0.110 (Z_{maxexttemp})$ |
| | | $\beta_{meanexttemp}Z_{meanexttemp} = 0.039 (Z_{meanexttemp})$ |
| | | $\beta_{exthumidity}Z_{exthumidity} = 0.192 (Z_{exthumidity})$ |
| | | $\beta_{pressure}Z_{pressure} = -0.007 (Z_{pressure})$ |
| | | $\beta_{rainfall}Z_{rainfall} = -0.007 (Z_{rainfall})$ |

| R Square | Adjusted R Square | *p*-Value |
|---|---|---|
| 0.470 | 0.470 | 0.001 |

| | | Legend |
|---|---|---|
| | | $Z'y_{h2}$ = Z score for Humidity 2 variable. |
| | | $\beta_{month}Z_{month} = 0.285 (Z_{month})$ |
| | | $\beta_{hour}Z_{hour} = -0.026 (Z_{hour})$ |
| | | $\beta_{position}Z_{position} = 0.568 (Z_{position})$ |
| **Humidity 2 (%), food area** | $Z'y_{h2} = 0.285(Z_{month}) - 0.026(Z_{hour}) + 0.568(Z_{position}) + 0.331(Z_{colony}) + 0.571(Z_{supers}) + 0.015(Z_{minexttemp}) + 0.061(Z_{maxexttemp}) + 0.028(Z_{meanexttemp}) + 0.170 (Z_{exthumidity}) + 0.024(Z_{pressure}) + 0.005(Z_{rainfall})$ | $\beta_{colony}Z_{colony} = 0.331 (Z_{colony})$ |
| | | $\beta_{supers}Z_{supers} = 0.571 (Z_{supers})$ |
| | | $\beta_{minexttemp}Z_{minexttemp} = 0.015 (Z_{minexttemp})$ |
| | | $\beta_{maxexttemp}Z_{maxexttemp} = 0.061 (Z_{maxexttemp})$ |
| | | $\beta_{meanexttemp}Z_{meanexttemp} = 0.028 (Z_{meanexttemp})$ |
| | | $\beta_{exthumidity}Z_{exthumidity} = 0.170 (Z_{exthumidity})$ |
| | | $\beta_{pressure}Z_{pressure} = 0.024 (Z_{pressure})$ |
| | | $\beta_{rainfall}Z_{rainfall} = 0.005 (Z_{rainfall})$ |

**Table 5.** *Cont.*

| R Square | Adjusted R Square | *p*-Value |
|---|---|---|
| 0.480 | 0.480 | 0.001 |

| | | |
|---|---|---|
| **Temperature 1 (°C), brood nest** | $Z'y_{t1} = -0.289(Z_{month}) + 0.121(Z_{hour}) + 0.436(Z_{position}) + 0.178(Z_{colony}) + 0.170(Z_{supers}) + 0.156(Z_{minexttemp}) + 0.180(Z_{maxexttemp}) - 0.135(Z_{meanexttemp}) - 0.025(Z_{exthumidity}) + 0.007(Z_{pressure}) - 0.014(Z_{rainfall})$ | $Z'y_{t1}$ = Z score for Temperature 1 variable. |
| | | $\beta_{month}Z_{month} = -0.289 (Z_{month})$ |
| | | $\beta_{hour}Z_{hour} = 0.121 (Z_{hour})$ |
| | | $\beta_{position}Z_{position} = 0.436 (Z_{position})$ |
| | | $\beta_{colony}Z_{colony} = 0.178 (Z_{colony})$ |
| | | $\beta_{supers}Z_{supers} = 0.170 (Z_{supers})$ |
| | | $\beta_{minexttemp}Z_{minexttemp} = 0.156 (Z_{minexttemp})$ |
| | | $\beta_{maxexttemp}Z_{maxexttemp} = 0.180 (Z_{maxexttemp})$ |
| | | $\beta_{meanexttemp}Z_{meanexttemp} = -0.135 (Z_{meanexttemp})$ |
| | | $\beta_{exthumidity}Z_{exthumidity} = -0.025 (Z_{exthumidity})$ |
| | | $\beta_{pressure}Z_{pressure} = 0.007 (Z_{pressure})$ |
| | | $\beta_{rainfall}Z_{rainfall} = -0.014 (Z_{rainfall})$ |

| R Square | Adjusted R Square | *p*-Value |
|---|---|---|
| 0.358 | 0.358 | 0.001 |

| | | |
|---|---|---|
| **Temperature 2 (°C), food area** | $Z'y_{t2} = -0.224(Z_{month}) + 0.280(Z_{hour}) + 0.206(Z_{position}) + 0.057(Z_{colony}) + 0.044(Z_{supers}) + 0.358(Z_{minexttemp}) - 0.024(Z_{exthumidity}) - 0.033(Z_{pressure}) - 0.037(Z_{rainfall})$ | $Z'y_{t2}$ = Z score for Temperature 2 variable. |
| | | $\beta_{month}Z_{month} = -0.224 (Z_{month})$ |
| | | $\beta_{hour}Z_{hour} = 0.280 (Z_{hour})$ |
| | | $\beta_{position}Z_{position} = 0.206 (Z_{position})$ |
| | | $\beta_{colony}Z_{colony} = 0.057 (Z_{colony})$ |
| | | $\beta_{supers}Z_{supers} = 0.044 (Z_{supers})$ |
| | | $\beta_{minexttemp}Z_{minexttemp} = 0.358 (Z_{minexttemp})$ |
| | | $\beta_{exthumidity}Z_{exthumidity} = -0.024 (Z_{exthumidity})$ |
| | | $\beta_{pressure}Z_{pressure} = -0.033 (Z_{pressure})$ |
| | | $\beta_{rainfall}Z_{rainfall} = -0.037 (Z_{rainfall})$ |

| R Square | Adjusted R Square | *p*-Value |
|---|---|---|
| 0.460 | 0.460 | 0.001 |

| | | |
|---|---|---|
| **Weight (kg)** | $Z'y_w = -0.302(Z_{month}) - 0.030(Z_{hour}) + 0.887(Z_{position}) + 0.349(Z_{colony}) + 0.963(Z_{supers}) + 0.032(Z_{maxexttemp}) - 0.005(Z_{exthumidity}) + 0.007(Z_{pressure})$ | $Z'y_w$ = Z score for Weight variable. |
| | | $\beta_{month}Z_{month} = 0.285 (Z_{month})$ |
| | | $\beta_{hour}Z_{hour} = -0.026 (Z_{hour})$ |
| | | $\beta_{position}Z_{position} = 0.568 (Z_{position})$ |
| | | $\beta_{colony}Z_{colony} = 0.331 (Z_{colony})$ |
| | | $\beta_{supers}Z_{supers} = 0.571 (Z_{supers})$ |
| | | $\beta_{maxexttemp}Z_{maxexttemp} = 0.061 (Z_{maxexttemp})$ |
| | | $\beta_{exthumidity}Z_{exthumidity} = 0.170 (Z_{exthumidity})$ |
| | | $\beta_{pressure}Z_{pressure} = 0.024 (Z_{pressure})$ |

| R Square | Adjusted R Square | *p*-Value |
|---|---|---|
| 0.743 | 0.743 | 0.001 |

Out of all significantly non-confounding factors, only the factors for which a significant linear relationship had been reported with any of the variables (brood nest, food area temperature, humidity and weight) were considered in the predictive regression equations. Table 5 reports the model summaries and standardized solutions for predictive regression equations for brood nest and food area humidity and temperature (humidity and temperature 1 and 2, respectively) and hive weight.

## 4. Discussion

Our results report average benchmark values of 51.7% ± 10.4 for humidity in the brood nest (Humidity 1), while these benchmark average values for the food area (periphery of the colony, Humidity 2) were 54.2% ± 11.7. Maximum values for humidity in the brood nest and food area were 91.3% and 92.6%, respectively, while minimum values in both cases were 15%. These benchmark values greatly differ from those generally reported in literature. However, authors such as Human et al. [61] reported information that may compare to our results. Similarities between ours and the study by Human et al. [61] could be ascribed to the similar climatic conditions between the Iberian and African subspecies [18].

In the context of the results reported by Chavez-Galarza et al. [18], certain environmental variables may act as selection driving agents. Hypothetically, genetically similar populations may be connected through the similar reaction patterns implemented by honeybees in response to those environmental selection agents, such as humidity. Still, these results should be considered cautiously, as the conditions in both studies may not be comparable. For instance, the study by Human et al. [61] was performed using three colonies during 4 days comprising a winter season which may account for a much lower humidity than in our case, in which relative humidity percentages are derived from the study of two complete years, hence comprising many very rainy periods. Conclusively, humidity benchmark values in literature for other subspecies may not be appropriate as reference values for some southern populations of Iberian honeybees.

Contextually, the conditions of development of bee larvae suggests that for egg hatching, a humidity of at least 55% may be necessary, with the highest levels of survival occurring with humidity ranging between 90% and 95% [62]. For instance, according to Doull [63], the optimal level for egg hatching must be set at 90–95% of relative humidity for the nest brood. The same author suggests that egg hatching may reduce by 68% when the relative humidity drops by 70%. In light of these ranges, some specific periods may have compromised egg hatching, although the effects of reduced humidity or drastic fluctuations may have been counteracted by honeybees as suggested in Figure 3. In the context of these results, our colonies may not have been able to develop normally in accordance to the minimum aforementioned necessary levels of humidity for proper larvae development provided in the bibliography [61–64]. As a result, the proper development of eggs and larvae would have been impeded.

Benchmark average temperature in the brood nest was 34.3 °C ± 1.5 (Temperature 1) while average temperature in the food area was 29.9 °C ± 5.8 (Temperature 2), with minimum and maximum temperatures ranging from 29.5 to 45.9 °C in the first case, and from 4.4 to 49.8 °C in the second, respectively. Our results are slightly over the benchmark values found in literature, as honeybees have been reported to maintain the temperature of the brood nest between 32 °C and optimally 35 °C to ensure the normal development of the brood. The inside temperature ranges from 33 °C to 36 °C in nest centers close to the worker brood [65]. Although this difference could be presumed to be very slight, research has shown that even small deviations (more than 0.5 °C) from the optimal brood temperatures have a significant influence on the development of the brood and health of the resulting adult bees [66]. This context suggests southern Iberian honeybees may need to develop an increased effort to maintain temperature within the aforementioned ranges compared to other populations, especially when the ambient temperature rise is above 35 °C.

Percentage of colony replacement in two whole years was 52%, due to mortality and weakness colonies. Switanek et al. [67] reported mortality rates of 16.7%, although they also suggest warmer and drier regions often accompany higher colony mortality rates which may support our findings. These authors attempted to isolate the influence of weather and climatic variability on honeybee mortality rates in Austria, although a multifactorial set of causes may be involved in colony loss. Contextually, numerous cases of winter losses and disorders continue to be recorded up to this date [68]. Despite this, an increase of 47.8% in honeybee colonies has occurred, many professional beekeepers have ceased activity, and some member states have experienced a decline in the number of bee colonies

by as much as 50% or more. These outputs may be ascribed to the effects of climate change (for instance, spring frost, drought, fires), certain chemical active substances and disturbances within the EU's internal market in honey. In this study, this ratio may also be ascribed to the occurrence of a failure to perform some typical management measures in beekeeping, such as the reinforcement of weak hives with breeding and the adult population from the strongest hives. As stated above, this measure was applied to intervene as little as possible in the normal development of the colonies. Consequently, episodes of robbing or infestation by *Galleria mellonella* larvae may occur. In addition, recent studies suggest that this measure may favor the most virulent strains of *Varroa destructor* [69].

When literature addresses colony loss it normally refers to losses during overwintering as suggested by Brodschneider et al. [70]. In the present study, losses correspond to the two full years that the study lasted for rather than to overwintering. Loss fractions of around 20% are commonly referred to in Central and northern Europe. However, in the area in which the study took place, the worst period is summer, which is long, very dry and registers very high temperatures that affect the colonies [25]. The years during which the study took place (2016 and 2017) are the ones reporting the hottest and driest registries since records exist [71], which may have implied heavier losses. For instance, losses during winter were 27% in 2016, with even higher loss rates in 2017

The annual cycle that bee colonies describe in southern Spain is characterized by two brood and workers peaks. First, a greater one in spring and a smaller one in autumn, which imply two other intermediate periods in which the populations descend (coinciding with summer and winter), reflected in the evolution of Weight (Figure 3). This distribution would be different from what is usually considered in Central and northern Europe, with a single brood and workers peak occurring in summer, with a reduction in populations occurring in winter.

Evidence of the capacity to thermoregulate and regulate humidity could be inferred across the results of the statistical tests performed in the study. For instance, no important differences between the variation coefficient of the relative humidity in the brood nest (20.092%) and food area (21.611%), for which environmental influence could be expected to be much greater at first. Coefficient of variation is a statistical measure of the dispersion of observations around the mean. This finding suggests Iberian bee colonies of this study were able to maintain internal beehive humidity at stable levels in comparison to the environment (CV of 44.463%), which compares and is consistent with the results from previous studies [8,20].

When Figure 2 was evaluated, a moderately high opposite correlation was found between environmental humidity and temperature in the brood nest and food area and vice versa. The magnitude of this opposite correlation was lower as we progressed towards the brood nest. In contrast, when external temperature (either mean, minimum or maximum) was compared to the temperature in the brood nest and food area, a direct progressively moderate to high correlation was reported as we progressed from the brood nest to the food area (closest to exterior). These findings may suggest a progressive reduction in the effects of environmental factors (temperature and humidity) on brood nest humidity and temperature, which may derive from thermoregulation and humidity regulation mechanisms implemented by honeybees.

Survivorship of honeybees has been reported to be strongly conditioned by high temperatures and, less relevantly, by relative humidity values, which may be in line with our results. In these regards Mardan and Kevan [72], reported a comfortable temperature spectrum which ranged between 26 and 36 °C, while increases to 45 °C and over implied the death of honeybees within 48 h. Contrastingly, higher relative humidity values have been suggested to improve survival.

Bastiaansen et al. [73] suggested two different alternatives when the behavior of honeybee colonies under episodes of mortality or diminished health was evaluated. One, the state in which the honeybee colony size falls beyond critical numbers, but remaining individuals manage to keep the colony's core temperature over acceptable thresholds. Two, a state in which such compensation is not feasible and the colony's nest temperature falls below critical thresholds, which dramatically promotes honeybee mortality leading to sudden death of the colony.

As opposed to healthy colonies, weak colonies may be characterized by low temperatures within the beehive which impede the ability of honeybees to seek for stored honey. Consequently, honeybees starve and die even if reserves are present in the beehive [20]. Events related to an increase in temperatures within closed beehives during transhumance during hot periods causes melting in the combs which ends in asphyxiating the bees [20].

In this context, genetic factors may be involved in improved thermal stability within the brood nest as according to Jones et al. [74]. Increased genetic diversity levels may determine honeybee workers temperature response thresholds, as highly diverse individuals may present an improved regulation ability of hive-ventilating behavior. This genetic variability at the individual level prevents exacerbated responses to temperature fluctuations at a colony level. This individualized regulatory behavior may be connected to the fact reported by some authors who have ascribed the lower or higher ability to endure high temperatures for a long time to the presence of heat shock proteins in honeybee larvae [75] and adults [76]. Additionally, thermal tolerance has been suggested to be influenced by inner bee related factors such as body size, among others [77].

When the hour factor was evaluated, Figure 2 suggests temperature and humidity in the brood nest and food area may increase during the day. Such increase became rather noticeable in closer areas to the outside of the hive. This suggested that hourly fluctuations in temperature may be buffered by honeybees as we progress towards the brood nest, while humidity levels may become independent from environmental humidity conditions as we progress towards the brood nest.

Environmental temperature and humidity linearly increased over the course of months through almost all the year. Temperature in the food area described almost the same trend as average, minimum and maximum environmental temperature as suggested by the high effect size and correlations. Temperature in the brood nest maintained a constant evolution over the year, provided the effect size of months on temperature in the brood nest presented the same magnitude than the correlation between temperature in the brood nest and mean, minimum and maximum environmental temperature. Our results are in line with those of Adam [78], who reported Iberian bees and North African bees to be resistant at temperatures when other bee subspecies would not venture forth.

Humidity in the brood nest and in the food area remained relatively constant as suggested by the similarities between the effect size of the month factor on each humidity and the correlation between environmental humidity and humidity in the brood nest and in the food area. A relative increasing pattern over the course of the months was described by environmental humidity. Fluctuations in environmental humidity were around 33.33% narrower than those in humidity in the food area and around 50% narrower than those in humidity in the brood nest (Figures 2 and 3). The data of this study indicate that Iberian honeybees (*A. m. iberiensis*) may put in less effort to regulate humidity within the hive, as well as possess a greater tolerance to low humidity conditions in comparison to the subspecies from other latitudes.

Our results may be supported by Milner's study [79], who provided early evidence for such differences by suggesting that Iberian bees may be unsuited to the tropical climate of South America, hence their failure to establish a feral population, North European bees adapted well to the harsher conditions that they found and feral colonies quickly established themselves over a wide area. Certainly, colonization by honeybees far outstripped that by settlers. For instance, in New Zealand and Tasmania, feral and managed colonies of *A. m. mellifera* have remained genetically isolated despite the massive importation of Italian bees.

When the temperature was assessed, our bee colonies presented a significantly greater ability for thermoregulation in the center of the brood nest with respect to peripheral areas, keeping it at values close to 34 °C across the two years that the study lasted (Figure 3). These results are similar to those published in literature for other subspecies [80]. In the same way, the results for weight, in general, also correspond to that which could be expected from normally developing colonies under a warm Mediterranean climate, with a few exceptions to be mentioned later in the text.

The trends described for temperature in the brood nest and in the food area suggest that conditioning factors may not significantly alter the capacity of Iberian honeybees to maintain stable temperature levels. Contrastingly, when humidity levels were assessed, Figures 2 and 3 report environmental humidity fluctuations of around 50% from October to December and from March to May can translate in fluctuations of 20% in relative humidity levels in the food area, and fluctuations of 10% in humidity levels in the brood nest.

The differences found concerning the capacity of the regulation of humidity and temperature can be explained by the greater adaptation of local bees to relatively wider ranges of humidity and drier climates than those in Central or northern Europe. This finding presents important practical implications, for instance, the European Food Safety Authority (EFSA) proposes that the ecotoxity tests of possible new molecules for agricultural use should be carried out in different regions of Europe [81], whose protocols are based on the recommendations of the European and Mediterranean Plant Protection Organization [82]. These recommendations normally use the only information available as a guideline, and such information is based on studies performed using Central European bees.

There are CROs (contract research organizations) across Europe conducting studies to enhance and implement such protocols. In this regard, our study aims to provide information as a way to adapt what is requested in the CROs for southern Europe and the real conditions in the area. This situation obviates the peculiarities of the populations from southern Europe; hence these protocols are not probably the most suitable to apply in research of these bees. Such differences should be taken into account as results provided by research may not be valid or relevant otherwise, but also at a practical level, as beekeepers may frequently acquire training based on information stemming from other climatic zones, what in turn derives in the application of inappropriate management policies, endangering the survival of local populations.

Among these incorrect policies, we could provide the example of the treatment against the *Varroa destructor*, whose application is regularly recommended during periods without brood, as autumn and winter. For Iberian colonies, such periods are most likely to take place in the summer, hence the application of the aforementioned application protocols, which may favor the prevalence of treatment resistance and promote the lack of effectiveness of the product. Under normal conditions, only one-third of the varroa mites are phoretic, so more than 60% of the population of Varroa is inside brood cells [83]. Moreover, Amitraz is currently the most effective, active synthesis substance in the treatment of varroa. However, Amitraz is not able to penetrate cells, hence authorized treatments include two brood cycles to maximize its effect. Contextually, in a colony lacking an operculated brood, 100% of mites will be susceptible to treatment, thus Amitraz's effectiveness will be high, and consequently, the likelihood of resistance will be reduced. Many factors can promote the inefficiency of the treatment, for instance, the application at undue times, inadequate doses, lack of rotation of substances, among others, but also, climatic conditions, beekeeping practices or even the environmental conditions within the hives [69].

A 13.5 kg higher beehive weight gain was reported when 2016 and 2017 were compared, which may also be ascribed to the rainy weather periods occurring during the spring months of 2017 [23], as the bee colony was experiencing a growth phase of population and reserves by that time. These differences are probably due to the confinement of the bees inside the hive, which could directly be a result of the prolonged rainy cycles in the spring of 2017, or to the presence of a natural predator—such as the European bee-eater (*Merops apiaster*), a protected bird that in days of adverse climatic conditions is concentrated in large numbers in the apiaries of the Iberian Peninsula—preventing the exit of foragers from the hives [84].

In any case, both situations may cause a reduction in the entry of food into the hive, so the colony may need to feed depending on stored resources, which could lead to a reduction in the potential growth of the colony and the profitability of beekeeping, especially if this situation lasts over time, as happened at the end of the spring of 2017. This situation is especially worrying if we consider the climate models for the Mediterranean area for the future years [21]. Increasingly long, hot and dry summers are expected, which may lead to more frequent and extensive food shortages in the

countryside, but also extreme rainfall events, which will prevent bees from collecting the food existing in the field. In the end, this will translate into the fact that the Iberian bee colonies become weakened and more sensitive to different diseases, whether viral, bacterial or fungal [85].

Under the current climate change situation, bioclimatic changes may occur in Central Europe, with a drift towards the conditions of the Mediterranean climate, which would have important consequences on the management of bee colonies, beekeeping productions and disease control in those countries. Therefore, all the information that is provided can be very valuable for the adaptation of European beekeeping to such new conditions.

Our results suggest (rather than just factors themselves) that a combination of climatic conditions and spatial configuration factors may determine a higher or lesser ability to regulate humidity within the hive. When the position factor was studied, we observed that the hives at the end (Figure 1) have a higher relative humidity and temperature inside the beehive, while those located closer to the center of the row appear to be drier and colder. This may be attributed to the fact that, in this case, the beehives at the end (1 to 5) were placed facing west (all hives faced south), which may have made them more exposed to the sun during the hottest hours (especially in summer), raising the average temperature, but also keeping them more exposed to the colder and wetter winds coming from the Atlantic Ocean, which are usually accompanied by rainfall, which may have increased humidity during the rainy seasons.

Regarding the presence of supers, the results are consistent with expectations, since the time in which they are placed is at the end of spring and early summer, so the presence of a single (1) or double (2) super, will coincide with a higher temperature and lower humidity. The increase in weight is also logical since supers are placed to allow the colony to reach its maximum size, accumulating the largest populations of bees, the largest amount of stored food and the largest area of combs (and therefore, the largest amount of wax). In short, the number of supers that the hive needs could be considered an indirect measure of the strength of the colony at the end of spring.

CATPCA reported five dimensions with significantly loaded components $\geq |0.5|$ (Table 3). Considering the distribution of factors significantly loading on each PC the clusters determined were as follows: "thermal sensation time variation" (PC1), "spatial location and configuration of the hive" (PC2), "annual cycle" (PC3), "weather depression moments" (PC4) and "barometric pressure influence" (PC5). These PCs comprise clusters of factors which may synergistically influence temperature and humidity levels within the hive, and consequently, the development of the colony. Still, the concrete effects of such factors or cluster of factors is undefined until further analyses are carried out. Among the methodological possibilities, CATREG may enable the study of these synergies.

For instance, Supplementary Tables S2 and S3 determinant coefficients or adjusted R squared values suggest predictive models may account for a moderate to high efficiency in predicting the variables of humidity and temperature in the brood and the food area (humidity and temperature 1 and 2, respectively) and hive weight.

Supplementary Tables S2 and S3 suggest that the most representative conditioning factors for all the variables were the factors of month, position within the row and the idiosyncrasies of the hive itself. A similar moderate linear relationship was reported between environmental humidity levels and humidity in the brood nest and in the food area. Contrastingly, the effects of rainfall and pressure were negatively linearly related to humidity in the brood nest while their effects were positively linear on humidity levels in the food area, which may provide evidence of the thermoregulatory ability for humidity within the hive, as when humidity levels decrease in the outside of the hive, the decrease may be somehow buffered in the food area to enable the humidity regulation processes occurring in the brood nest. The same situation was reported for temperature in the brood nest and the food area as suggested by the β standardized coefficients of each respective variable. However, the thermoregulatory capacity may be quantifiably higher than humidity regulatory capacity as suggested by the lower magnitude of the β standardized coefficients for average environmental temperature. As the number of supers

increases, their relevancy on humidity in the food area and temperature in the brood nest increases, which may account for the certain effect of their addition on thermoregulatory capacity.

The information that these regression models report may provide us with a better understanding on how honeybee populations react to these factors or how they become affected by them. Contextually, this may open the door for future studies that seek a deeper exploration of the biology of honeybees, their health and productivity, providing beekeepers with more efficient tools to improve the profitability of beekeeping farms, as well as helping their conservation actions. Additionally, the regression equations designed in the present study provide an important tool to improve the existing knowledge of bee colonies, specifically the southern ones of Iberian Peninsula. These equations are based on large amounts of data and can be used to make computer models that allow us to predict the behavior and evolution of bee populations, or improve the existing ones [86]. This is essential, as long as regression studies are based on a sufficient amount of data, to predict the effects over climate change, the use of chemicals in intensive agriculture or the reduction in biodiversity, especially plants, which may become progressively extinct at a much higher rate than vertebrates [87].

The use of objective data derived from hive monitorization may be the starting point for the revolution of the knowledge available on beehives and honeybees themselves. Through the use of hive integrated sensors, videographic advances and specific software, not only will researchers be able to integrate the findings derived from the present paper but also be able to obtain accurate and objective data of a diverse indole (for instance, perform colony and brood assessments and monitor flight activity) with the ability to involve much lower personnel and time demands minimizing beehive manipulation and handling [88,89]. The possibility to obtain timely observations helps to improve the quality of predictive models, which may be profitable, even in the context of relatively high initial investments. Furthermore, as suggested by Bermig et al. [88], new software and technological advances can help to save drawbacks such as those derived from the need to maintain sufficient sample sizes to maintain statistical power, which in turn, determines the ability to model present data and to predict future data.

New technological advances will offer the possibility to delve into the biology of bees in the Iberian Peninsula, with the help of new automated photographic analysis systems of the surface of food reserves and broods inside the colony [88], as well as the bee entrance to and exit from the hive [82] to provide data which may enable the building of predictive models. These issues, together with a greater knowledge of the pathology of bees, will allow us to make better decisions, both at the beekeeper and the legislative level, to help protect these insects and ensure that they remain one of the fundamental elements for the maintenance of the biodiversity of our planet.

## 5. Conclusions

The bees of the Iberian subspecies used in this study have a low regulatory ability in relative humidity within the colony compared with temperature, as well as a great tolerance to low humidity conditions. The colonies presented a greater ability for thermoregulation in the center of the brood nest with respect to peripheral areas, keeping it at values close to 34 °C. The prolonged rainy cycle of the spring of 2017 caused a reduction in the potential growth of colonies and the profitability of beekeeping in southern Spain. Real-time monitoring has proved to be a very useful and effective tool to control the conditions inside the hive and to assess their relationship with climatic factors.

**Supplementary Materials:** The following are available online at http://www.mdpi.com/2071-1050/12/16/6431/s1, Table S1. Summary of the significant (*p* < 0.05) pairwise differences (Green) obtained after Dunn's test for the levels of the factors of Year, Month, Hour, Position, Hive, Supers, and Wind (n = 545,122) assessed in Iberian bee colonies; Table S2. Medians for internal environmental hive and hive weight variables for each of the levels of the factors of Year, Month, Hour, Position, Hive, Supers, and Wind (n = 545,122) assessed in Iberian bee colonies (the greener the higher, the redder the lower); Table S3. Summary of the results of CATREG analysis for Humidity 1 (brood nest) and 2 (feed area), Temperature 1 (brood nest) and 2 (feed area) and weight.

**Author Contributions:** Conceptualization, J.M.F.S. and S.G.-L.; methodology, S.G.-L.; software, F.J.Q.L.; validation, J.M.F.S., V.G.L. and F.J.N.G.; formal analysis, F.J.N.G.; investigation, V.G.L. and S.G.-L.; resources, J.M.F.S.; data curation, S.G.-L.; writing—original draft preparation, S.G.-L.; writing—review and editing, F.J.N.G.; visualization, S.G.-L.; supervision, F.J.N.G.; project administration, J.M.F.S.; funding acquisition, J.M.F.S. and F.J.Q.L. All authors have read and agreed to the published version of the manuscript.

**Funding:** This research was funded by the European Union's European Regional Development Fund (ERDF) 2014–2020, through the National Institute for Agricultural and Food Research and Technology (INIA) of Spain, and the projects "Holistic evaluation of risk factors in honeybees and wild pollinators. The situation in Spain", grant number RTA2013-00042-C10, and "Improvement of production conditions in beekeeping: development of wax decontamination processes, impact of veterinary treatments in conventional and ecological management, and wellness indicators of the colony", grant number RTA2017-00058-C04-04.

**Acknowledgments:** We would like to thank all the staff of the Departments of Zoology and Computer Architecture, Electronics and Electronic Technology who have collaborated in this study, whose work was essential for its development.

**Conflicts of Interest:** The authors declare no conflict of interest.

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
