# Peer review of "Regulation of Microclimatic Conditions inside Native Beehives and Its Relationship with Climate in Southern Spain"

_sustainability, doi:10.3390/su12166431_

Round 1

Reviewer 1 Report

This manuscript provides data and statistical analyses pertaining to the long term viability of the honey bee subspecies native to the Iberian penninsula, Apis mellifera iberiensis. Climate change is likely to threaten this subspecies, especially if it cannot maintain the hive temperature and humidity ranges it requires. The measurement and analyses of hive temperature, humidity, weight and environmental variables over about two years are certainly appropriate to this issue. This study can help researchers develop plans to conserve this subspecies in Iberia. The methods can be used also to study honey bee colonies elsewhere in the world.

     An important limitation to the study is that only one temperature probe was used in the honey super and one in the brood nest for each hive. Similarly, one humidity probe was installed in the honey super and one in the brood nest. Numerous other studies have shown that temperature is not uniform throughout the hive. This is especially important in winter when the bees cluster as outdoor temperatures drop below 10 degrees C. At 0 degrees they are tightly clustered. The bees warm their cluster but not the rest of the hive. If the bees cluster away from the probe, temperature measurements will not be helpful. Clustering at one side of the hive in winter is very common. Fig. 3 shows that winter temperatures often fell below 10 degrees. Humidity will also be different inside the cluster, compared to the rest of the hive. I feel that the authors should eliminate the data for the winter months and use only the data for the times that outdoor temperatures exceeded 10 degrees. Future studies should include many probes placed throughout each hive.

     Other comments:

The title is much too long – 67 syllables! It could easily be shortened to half this length or less.

Line 25: Acronyms (CATPCA and CATREG) should be spelled out and explained on first use in the manuscript. We don’t get that CATREG until lines 255 and 256.

Line 135: Use “harvest” in place of “obtention”

Line 170: What is “posterior”? Should the word be “researcher”?

Line 230: should be spelled “through”

Line 243: should be spelled “analysis”

Results: The methods do not justify reporting mean values to 5 significant figures. Use 3 significant figures. For example in line 369, the mean should be reported as 51.7% + 10.4

Author Response

This manuscript provides data and statistical analyses pertaining to the long term viability of the honey bee subspecies native to the Iberian penninsula, Apis mellifera iberiensis. Climate change is likely to threaten this subspecies, especially if it cannot maintain the hive temperature and humidity ranges it requires. The measurement and analyses of hive temperature, humidity, weight and environmental variables over about two years are certainly appropriate to this issue. This study can help researchers develop plans to conserve this subspecies in Iberia. The methods can be used also to study honey bee colonies elsewhere in the world.

     An important limitation to the study is that only one temperature probe was used in the honey super and one in the brood nest for each hive. Similarly, one humidity probe was installed in the honey super and one in the brood nest. Numerous other studies have shown that temperature is not uniform throughout the hive. This is especially important in winter when the bees cluster as outdoor temperatures drop below 10 degrees C. At 0 degrees they are tightly clustered. The bees warm their cluster but not the rest of the hive. If the bees cluster away from the probe, temperature measurements will not be helpful. Clustering at one side of the hive in winter is very common. Fig. 3 shows that winter temperatures often fell below 10 degrees. Humidity will also be different inside the cluster, compared to the rest of the hive. I feel that the authors should eliminate the data for the winter months and use only the data for the times that outdoor temperatures exceeded 10 degrees. Future studies should include many probes placed throughout each hive.

Response: We are aware of this evolution in the colony; however, we have found that during the process of "drafting" there was a misunderstanding. Initially "the sensor was always within its area (either breeding or feeding)" has been mistaken with "Sensors were located at the same locations inside the hives". We've fixed it as: "A set of sensors controlled by a Waspmote®-based processor is placed within each beehive, and periodic reviews were performed to ensure that each sensor was within the proper area within the hive, repositioning sensors in case it was necessary".

On the other hand, only the minimum daily ambient temperatures normally drop below 10oC, while temperatures inside the hives did not drop beyond 20 oC. This explains why bees in our area do not cluster during the winter and maintain breeding areas along the winter. Not so in summer, in which during the months of July and August it is frequent not to find anything of breeding inside the hive.    

Other comments:

The title is much too long – 67 syllables! It could easily be shortened to half this length or less.

Response: Changed

Line 25: Acronyms (CATPCA and CATREG) should be spelled out and explained on first use in the manuscript. We don’t get that CATREG until lines 255 and 256.

Response: Included

Line 135: Use “harvest” in place of “obtention”

Response: Changed

Line 170: What is “posterior”? Should the word be “researcher”?

Response: Changed to later

Line 230: should be spelled “through”

Response: Corrected

Line 243: should be spelled “analysis”

Response: Corrected

Results: The methods do not justify reporting mean values to 5 significant figures. Use 3 significant figures. For example, in line 369, the mean should be reported as 51.7% + 10.4

Response: Corrected

Reviewer 2 Report

The study of monitoring the temperature and humidity conditions of the Iberian honeybee colonies is very interesting and important for apiculture and also for science. Regression models may be also important in predicting the effects of environmental conditions on honey bee colonies development and production. Majority of the manuscript is dedicated to the development recording and data analyses, which is not my prime expertise. The chapters of the manuscript Introduction, M&M, Results and Discussion need to be shortened and condensed that the subject  will easier to follow and be clearly presented.   I have added some comments and suggestions from apicultural point of view. I am not able to properly evaluated statistical analyses and I advise an expert in that topic.

Author Response

The study of monitoring the temperature and humidity conditions of the Iberian honeybee colonies is very interesting and important for apiculture and also for science. Regression models may be also important in predicting the effects of environmental conditions on honey bee colonies development and production. Majority of the manuscript is dedicated to the development recording and data analyses, which is not my prime expertise. The chapters of the manuscript Introduction, M&M, Results and Discussion need to be shortened and condensed that the subject will easier to follow and be clearly presented.   I have added some comments and suggestions from apicultural point of view. I am not able to properly evaluated statistical analyses and I advise an expert in that topic.

Response: We thank the reviewer for the kind comments and will address his/her suggestions as we possibly can.

16: Describe more data, findings and conclusions in the Abstract; instead of long introduction and general information. 

Response: We have restructured the section following your suggestions.

101: That are environmental conditions. Climate factors affect wider geographical areas.

Response: We changed nomenclature following reviewer’s suggestions.

119-120: Why this happened? It seems that there is a not correct procedure for hives distribution. Please explain.

Response: At least in Spain, this is the usual way in commercial beekeeping to place the hives (in one or two rows), in order to facilitate handling.

2.5. Sub - chapter is very long described and difficult to follow. In some parts it describes and comprises material & methods, literature data with results and discussion. That way of description, data analyses may have been self-sufficient topic for separate paper in this matter.   

Response: We summarized statistical analyses, while trying to keep the whole meaning of what was statistically performed in order not to confuse readers. We left the information in regards the test that were performed and how to interpret the results. The rest was deleted or relocated in a different section.

491-492: Similarities between subspecies in different experiments could not be compared. For that kind of comparison, different subspecies need to be tested in the same environment using same monitoring technology.

Response: We agree, it is precisely in the text that we argue that comparison was not feasible. In any case, we have reformulated this sentence, and we have also tried to avoid making references to subspecies, and only to local bee populations regarding bibliography to avoid confusion.

505-507: Sensors were located at the same locations inside the hives. Therefore, they monitored different developmental stages of bee (from eggs to capped brood). Differences in humidity is not only reflection of environmental conditions, but also reflection of colonies development. It seems that all these data were not recorded accordingly and there may be significant differences between colonies in the same environment. These aspects need to be discussed and also considered in M&M and Results section. Similar phenomenon can appear when measured temperature and humidity.  

Response: We are aware of this evolution in the colony; however, we have found that during the process of "drafting" there was a misunderstanding. Initially "the sensor was always within its area (either breeding or feeding)" has been mistaken with "Sensors were located at the same locations inside the hives". We've fixed it as: "A set of sensors controlled by a Waspmote®-based processor is placed within each beehive, and periodic reviews were performed to ensure that each sensor was within the proper area within the hive, repositioning sensors in case it was necessary".

557-558: Is there any correlation or explanation between temperature and humidity between healthy colonies and weak colonies before dying? The colonies mortality was high and that data may also have an impact on total temperature and humidity distribution inside and between colonies.

Response: It could be. But such an approach would require the inclusion of a wide variety of factors, which would greatly increase the length of the article and depart from the objective of the article.

Reviewer 3 Report

This manuscript investigates temperature and humidity recorded from inhive as well as from surrounding sensors using the “Iberian honey bee” Apis mellifera iberiensis. This data should serve as benchmark to investigate further developmental changes and/or adaptations from external influences such as the climate change on this particular bee sub-species. As a main result, the authors present that Iberian bees are limited in regulating humidity within the colony when compared to temperature.

The manuscript is well written and understandable in most parts. However, I fail to understand what knowledge gap this study addresses. One has to consider that the Iberian honey bee, as the authors introduce them, are a very complex and inhomogeneous sub-species. The Atlantic side of the Iberian Peninsula alone, hosts 16 different haplotypes (see Pinto et al. 2012, Apidologie) and comprises two of four linages (A and M). The distribution of haplotypes of this sub-species throughout the region is very inhomogeneous with a gradient present from southwest to northeast. The frequency of African haplo-types decreases as the West European haplo-types increases and vice versa (see Canovas et al. 2010, Apidologie). The Iberian Peninsula has the highest genetic diversity and complexity in honey bees across Europe. There is no definite geographic structure of Iberian honeybee populations (see Canovas et al. 2010, Apidologie). Therefore, it is more than questionable that the data you provide from only 10 Iberian colonies from only one location is representative for the whole sub-species. This is by the way nowhere mentioned in the manuscript.

Further, it is remarkable that most of your studied colonies died within the two-year period despite management practices and Varroa treatment. As a beekeeper myself, I know that it is not easy sometimes to act on all fronts. Especially fighting Varroa might be a challenge. Nevertheless, loss rates above 20% are not acceptable and a sign of poor management - who knows in which way this has influenced study results. Altogether, considering the above-mentioned flaws, I must advise the editor to reject this manuscript.

Author Response

Comments and Suggestions for Authors

This manuscript investigates temperature and humidity recorded from inhive as well as from surrounding sensors using the “Iberian honey bee” Apis mellifera iberiensis. This data should serve as benchmark to investigate further developmental changes and/or adaptations from external influences such as the climate change on this particular bee sub-species. As a main result, the authors present that Iberian bees are limited in regulating humidity within the colony when compared to temperature.

The manuscript is well written and understandable in most parts. However, I fail to understand what knowledge gap this study addresses. One has to consider that the Iberian honey bee, as the authors introduce them, are a very complex and inhomogeneous sub-species. The Atlantic side of the Iberian Peninsula alone, hosts 16 different haplotypes (see Pinto et al. 2012, Apidologie) and comprises two of four linages (A and M). The distribution of haplotypes of this sub-species throughout the region is very inhomogeneous with a gradient present from southwest to northeast. The frequency of African haplo-types decreases as the West European haplo-types increases and vice versa (see Canovas et al. 2010, Apidologie). The Iberian Peninsula has the highest genetic diversity and complexity in honey bees across Europe. There is no definite geographic structure of Iberian honeybee populations (see Canovas et al. 2010, Apidologie). Therefore, it is more than questionable that the data you provide from only 10 Iberian colonies from only one location is representative for the whole sub-species. This is by the way nowhere mentioned in the manuscript.

Response: We understand your concerns and agree with your suggestions. Although, provided the genetic diversity of the Iberian subspecies stating our results may apply to the entire subspecies could be adventurous, the data collected highlights some insights in terms of the existing and traditionally accepted literature. Therefore, we think that this data could complement the knowledge of the bee of our region (at least that of the south of the Iberian Peninsula), as certain protocols and regulations developed at European level may not fit the conditions of the area properly which particular conditions may be misrepresented. Following your suggestions, we have reorganized the article so that it focuses on the local bee population, rather than the Iberian subspecies. The fact that this study has been developed in the warmest area of Europe may be worth noting [1], Hence the climatic conditions that our bees faced may be similar to those to be experienced in larger areas of Europe in the coming decades.

  1. IBERIAN CLIMATE ATLAS. AIR TEMPERATURE AND PRECIPITATION (1971-2000); Ed. Agencia Estatal de Meteorología (Ministerio de Medio Ambiente y Medio Rural y Marino de España) and Instituto de Meteorologia de Portugal, 2011. http://www.aemet.es/documentos/es/conocermas/publicaciones/Atlas-climatologico/Atlas.pdf

Further, it is remarkable that most of your studied colonies died within the two-year period despite management practices and Varroa treatment. As a beekeeper myself, I know that it is not easy sometimes to act on all fronts. Especially fighting Varroa might be a challenge. Nevertheless, loss rates above 20% are not acceptable and a sign of poor management - who knows in which way this has influenced study results. Altogether, considering the above-mentioned flaws, I must advise the editor to reject this manuscript.

Response: Usual practices in beekeeping such as the reinforcement of weak colonies with breeding and adult population of the strongest colonies, were not performed to fulfill the non-interventionist condtions of the study. This, along with the especially adverse summer conditions in the Cordoba area, may explain the high replacement rate. Despite this, our loss data is similar or even lower than those that have been reported in the press by organizations and commercial beekeepers in our area.

Round 2

Reviewer 1 Report

This manuscript is much improved and should be published.  

Lines 484-485 contain a Spanish phrase "condiciones climaticas similares" that should be translated to English.

Author Response

This manuscript is much improved and should be published. 

Response: We thank the reviewer for his/her kind comments.

Lines 484-485 contain a Spanish phrase "condiciones climaticas similares" that should be translated to English.

Response: Corrected.    

Reviewer 2 Report

Authors responded adequately on my comments. There are still some flaws in the manuscript, as repetitions of references indicated in the manuscript (line 481 and etc).  

Author Response

Reviewer 2

Authors responded adequately on my comments. There are still some flaws in the manuscript, as repetitions of references indicated in the manuscript (line 481 and etc). 

Response: We followed reviewer’s suggestions and corrected citation repetitions.

Reviewer 3 Report

In light of the substantial changes following my objection on the generalization of Iberian sub-species, I find justification of high mortality rates in a non-interventionist approach. Authors, too, have discussed this appropriately.

Two minor comments, that I would like to suggest, are as follows:

  • Reviewer #1 stated on the first version of the manuscript “L557-558: Is there any correlation or explanation between temperature and humidity between healthy colonies and weak colonies before dying? The colonies mortality was high and that data may also have an impact on total temperature and humidity distribution inside and between colonies.” – I fully agree to this comment. Even if an additional analysis is not feasible, this point must be discussed. Please consider this in your discussion section, respectively.
  • L711-728: Please also include a few lines to state what is necessary to generate these data for prediction modelling, i.e. not only temperature and humidity data, but also technical advances to digitize colony and brood assessments and monitor flight activity (Colin et al. 2018, Bermig et al. 2020). This could be of high relevance to further improve regulatory relevant methodologies.

Please review and include these two references:

Colin, T.; Bruce, J.; Meikle, W.G.; Barron, A.B. The development of honey bee colonies assessed using a new semi-automated brood counting method: CombCount. PLoS One 2018, 13, e0205816, doi:10.1371/journal.pone.0205816.

Bermig, S.; Odemer, R.; Gombert, A.; Frommberger, M.; Rosenquist, R.; Pistorius, J. Experimental validation of an electronic counting device to determine flight activity of honey bees (Apis mellifera L.). J. Cultiv. Plants 2020, 72, 132–140, doi:10.5073/JfK.2020.05.03.

Author Response

In light of the substantial changes following my objection on the generalization of Iberian sub-species, I find justification of high mortality rates in a non-interventionist approach. Authors, too, have discussed this appropriately.

Response: We thank reviewer for his/her kind comments.

Two minor comments, that I would like to suggest, are as follows:

Reviewer #1 stated on the first version of the manuscript “L557-558: Is there any correlation or explanation between temperature and humidity between healthy colonies and weak colonies before dying? The colonies mortality was high and that data may also have an impact on total temperature and humidity distribution inside and between colonies.” – I fully agree to this comment. Even if an additional analysis is not feasible, this point must be discussed. Please consider this in your discussion section, respectively.

Response: The reviewer suggesting was included in the discussion section.

L711-728: Please also include a few lines to state what is necessary to generate these data for prediction modelling, i.e. not only temperature and humidity data, but also technical advances to digitize colony and brood assessments and monitor flight activity (Colin et al. 2018, Bermig et al. 2020). This could be of high relevance to further improve regulatory relevant methodologies.

Response: Information requested by reviewer was included.

Please review and include these two references:

Colin, T.; Bruce, J.; Meikle, W.G.; Barron, A.B. The development of honey bee colonies assessed using a new semi-automated brood counting method: CombCount. PLoS One 2018, 13, e0205816, doi:10.1371/journal.pone.0205816.

Bermig, S.; Odemer, R.; Gombert, A.; Frommberger, M.; Rosenquist, R.; Pistorius, J. Experimental validation of an electronic counting device to determine flight activity of honey bees (Apis mellifera L.). J. Cultiv. Plants 2020, 72, 132–140, doi:10.5073/JfK.2020.05.03.

Response: References were reviewed and included as suggested.